# Relationship between Smoking, Physical Activity, Screen Time, and Quality of Life among Adolescents

**DOI:** 10.3390/ijerph17218043

**Published:** 2020-10-31

**Authors:** Xiaosheng Dong, Meng Ding, Wenxin Chen, Zongyu Liu, Xiangren Yi

**Affiliations:** 1Department of Sport and Health, School of Physical Education, Shandong University, Jinan 250061, China; dxiaosheng@hotmail.com (X.D.); wenxinchen@sdu.edu.cn (W.C.); 201700292037@mail.sdu.edu.cn (Z.L.); 2College of Physical Education, Shandong Normal University, Jinan 250014, China; dingmeng@sdnu.edu.cn

**Keywords:** adolescent, quality of life, smoking, physical activity, screen time, China

## Abstract

Background: Quality of life (QOL) is a crucial part of evaluating health conditions IN adolescents. The purposes of this study were to (1) examine the relationship of QOL and smoking, physical activity (PA) and screen time (ST) among Chinese adolescents, (2) explore the relationship between PA-ST combination and QOL of adolescents, and (3) investigate the dose-response relationship between PA-ST and QOL. Methods: This study randomly selected 12,900 adolescents (11–18 years) from 13 administrative regions in Shandong Province, China. The data gathering tools for Smoking (tobacco), PA (PAQ-A) and ST (average daily time for ST) and QOL questionnaire (child and adolescent quality of life scale) were completed among all adolescents. Statistical analysis was performed by T test, chi-square test and multiple linear regression. Results: 12,641 adolescents (aged 12–18) completed the study. In multiple linear regression models, the result demonstrated that the adolescents from rural areas, with high ST, low PA, and smoking, with older age and low socioeconomic status, showed a lower QOL score. First-time smokers under 10 years revealed the lowest QOL, and PA > 30 min five days per week have the highest QOL. In addition, boys and girls with PA > 30 min three to four days per week in high ST group obtain the higher scores (boys β = 5.951, girls β = 3.699) than low PA-low ST groups. Conclusions: Adolescents from rural areas suffer from a relatively poorer QOL. More than 30 min of PA five or more days for boys and three or four days per week for girls could decrease negative effects of ST and improve QOL.

## 1. Introduction

Quality of life (QOL) has become a critical measure of outcome for people with mental and physical health concerns [1], and identifies different health levels of adolescents [2] in multiple dimensions such as physiological, psychological, emotional and social [3]. With the advancement of medicine, health is increasingly involved in the physical, psychological and social fields [4], but the current disease-based health services ignore several health problems of adolescents [5]. Therefore, it is of great significance to discuss the QOL of adolescents.

It is known that social cognitive theory (SCT) [6] is a frequently-used way to illustrate and alter quality of life [7]. SCT takes the interactions among personal, social and environmental factors to interpret and predict behaviors [6]. The effects of a number of factors on the quality of life of adolescents have been studied such as age, gender [8], place of residence [9], socioeconomic status [10] etc., as well as the correlation between adolescents’ PA, ST, smoking and QOL [11,12,13].

Notably, tobacco use has become the second leading death risk factor in the world [14], and a have shown that China has about 301 million smokers (45.5% of the world’s smokers), and about 1 million deaths each year are related to tobacco [15]. As smoking attracts academic attention [16,17,18], it is an urgent situation that the prevalence of smoking among people aged 15–24 years has gone up from 8.3% in 2003 (8.0–8.6 years) to 12.5% in 2013 (12.2–12.9 years) [19], which may endanger the health of Chinese adolescents. Previous researches illustrated that low PA levels in children and adolescents is associated with unhealthy states such as obesity, cardiometabolic health risks and mental health problems [20,21]. Nearly 80% of the world’s young people do not meet the World Health Organization recommended adolescent PA guidelines [22] and in China this figure is 70% [23]. Some studies have indicated that high ST is associated with obesity [24] and various non-communicable diseases [25]. However, accompanied by technology advances, screen time for teens watching TV or playing with a mobile phone has become an important part of daily life [26]. This trend of low PA and high ST may endanger the health of adolescents. Therefore, controlling smoking problems, reducing ST, and increasing PA has become the focus of public health.

Optimal PA could relieve the negative effects of high ST on QOL [27]. Some studies have pointed out that even if the PA guidance recommendations are met, the risk of psychological distress will increase if the guidelines on ST are exceeded [28]. In addition, many studies reported that there are gender differences in quality of life [29], smoking [30], PA [31], and ST [32] among adolescents. More specifically, compared to males, females had lower quality of life scores, smoking rates, and physical activity in most of the dimensions. The findings indicate that boys and girls watch the same amount of TV, and boys use more electronic games/computers than girls [32]. Therefore, we need to formulate corresponding prevention and improvement strategies, and target the differences in PA, ST and QOL between male and female teenagers.

Although relevant studies have explored the influence of smoking, PA, and ST on the QOL of adolescents, there are several problems. First, the correlation between smoking status and self-reported quality of life are unclear [33]. The results demonstrated that students who met the recommended physical activity goals had a mixed quality of life [34]. Some studies have shown that ST has little correlation with quality of life [35,36]. Therefore, the correlation between smoking, PA, ST, and quality of life is still debatable. Besides, the dose-response relationship between the combination of ST and PA and QOL has not been studied. Secondly, it is known that environmental factors are also related with QOL [37]. Previous studies showed that limited family material resources and lower social status also damaged QOL in adolescence [10], but there are few studies on the relationship between place of residence and QOL among Chinese adolescents. Thirdly, there is a lack of large sample studies to explore the effects of smoking, physical activity and screen time on the QOL of Chinese adolescents, while accounting for distinct demographic factors. The purposes of this study are to (1) explore the relationship of QOL and smoking, PA and ST among Chinese adolescents; (2) explore the relationship between PA-ST combination and the QOL of Chinese adolescents, and investigate the dose-response relationship of QOL and combination of PA for over 30 min per week and high/low ST, while accounting for demographic factors such as age, sex, place of residence and socioeconomic status. We hypothesized that smoking, physical activity and screen time are related to the quality of life of Chinese adolescents, and high physical activity can improve the negative impact of high screen time on quality of life. This research is expected to pave the way for formulating strategies to improve the QOL of Chinese adolescents.

## 2. Materials and Methods

### 2.1. Participants and Study Design

This research used a cross-sectional study design to investigate adolescents in Shandong Province, China from 2017 to 2018. There are 90 public middle and high schools that were randomly selected from 13 administrative regions in terms of the province’s [38] specific geography, population and socio-economic level. Three high schools and seven middle schools were selected randomly in each region consisting of at least 300 students in each school, and 100 students in each grade, respectively. A total of 12,900 young people (11–18 years) were chosen for this study, but only 12,641 persons participated because 259 adolescents (132 boys and 127 girls) dropped out. Parents and students filled out a consent form prior to enrollment in this survey. This study has been approved by the Ethics Committee of Shandong University (20180517).

Ninety physical education (PE) teachers were recruited from middle and high schools who had previous experience in evaluating youth fitness programs. Workshops were held to train PE teachers and graduate students who administered all questionnaires by following standard operating procedures. All PE teachers had to participate in two training seminars that facilitated quality control for the method of assessment. Students completed a 40 min questionnaire independently during PE class under the guidance of PE teachers and graduate students. It is well-noted by all participants that all the data was collected on a voluntary, anonymous and confidential basis, was kept on a password-protected website, and only direct researchers could access the data. The sample questionnaires are published in Supplementary Materials.

### 2.2. Assessments

Demographic factors: These include gender, age, grade, place of residence and socioeconomic status. Two options for place of residence are urban and rural areas. The SES of guardians was investigated from the aspects of educational background and occupational status [39,40].

Physical Activities: Physical activity (PA) was evaluated by the physical activity questionnaire for adolescents (PAQ-A). This scale is a revised version of the Physical Activity Questionnaire for Children (PAQ-C) and aims to assess the PA level of adolescents [41], and its effectiveness and reliability with the Chinese youth population have been verified [42]. Reliability of the questionnaire was analyzed by Cronbach’s alpha (α = 0.732). The questionnaire asks young people what they have done during most of their breaks in the past seven days. It applies a 5-point system (1–5) and the higher score indicates the higher PA level. The results can be divided into low PA level (1–1.9 points) and high PA level (2–5 points) [43]. In addition, we investigated how many days adolescents had exercised for more than 30 min in the past 7 days (including all kinds of exercises that increase heart rate and make people breathe faster), and the options were divided into 0 days, 1–2 days, 3–4 days or ≥5 days.

Screen time: The adolescents who participated in this study all reported the average daily time for watching TV or video, using computers, video games, tablets PC, mobile phones or other social tools, as well as the average time spent using computers for tasks unrelated to learning, and according to the recommendations of international ST this figure was divided into high ST (≥2 h per day) and low ST (<2 h per day) [44].

Smoking (tobacco): Four questions were applied to determine the smoking status of teenagers: (1) Have you ever tried smoking, even if it was a puff? (2) How old were you when you first tried cigarette smoking, even if it was a puff? (3) How many days did you smoke in the last 30 days? (4) How many cigarettes did you smoke per day on smoking days? According to question (1) (2), the age of first smoking is divided into four groups: no, age 10 and under, age 11–14, age 15 and above; According to question (3) (4), the number of cigarettes per month was calculated, and then divided into three groups: never smokers, occasional smokers (smokes 5 or under cigarettes per month), and regular smokers (smokes 6 or more cigarettes per month).

Quality of Life: The quality of life was assessed by the Quality of Life Scale for Child and Adolescents [45]. Reliability of the questionnaire was analyzed by Cronbach’s alpha (α = 0.892). Social and psychological function (α = 0.809), physiological and mental health (α = 0.703), and living environment (α = 0.784), had reliability coefficients higher than 0.7. Factor analysis was performed to evaluate, construct validity using the Kaiser-Meyer-Olkin (KMO = 0.866, *p* = 0.00). Using the Chinese version, Cronbach’s reliability α = 0.885, content validity (r = 0.63, *p* < 0.01), and construct validity (KMO = 0.890, *p* = 0.00) were carried out, which have good reliability and validity, meet the requirements of psychometrics [45], and has been used to investigate QOL [46].

The scale contains three factors. Social and psychological function factors mainly deal with teacher-student relationship, peer relationship, parent-child relationship, learning ability and attitude, self-concept and other dimensions; physiological and mental health factors mainly cover somatic sensation, negative emotions, work attitude, etc.; the living environment factor mainly controls the dimensions of life convenience, activity opportunity, exercise ability, etc. A higher score indicates a better QOL. The scale assesses the QOL of children and adolescents from four aspects: life, psychology, social function, and living environment.

### 2.3. Statistical Analysis

Descriptive statistics were performed on all variables by chi-square test and T test or analysis of variance (ANOVA) according to gender. Continuous variables were expressed as mean and standard deviation (mean ± standard deviation), and categorical variables were expressed as numbers (*n*) and percentages (%) Effect sizes (Cohen’s d) were categorized as small if d > 0.2, medium if d > 0.5 and large if d > 0.8 [47]. Multiple linear regression was used to analyze the relationship between QOL and adolescent smoking, PA, ST, and PA-ST combination. The model was adjusted for age, place of residence and socioeconomic status, in addition to adjustment for smoking (regarding ST and PA), physical activity (regarding ST and smoking) or screen time (regarding PA and smoking). Potential multicollinearity among the predictor variables was assessed by examining the variance inflation factors’ (VIF) diagnostic statistics. Although there is no formal VIF value for detecting multicollinearity, values of VIF 10 or above are often indicative of multicollinearity [48]. The unstandardized coefficients with a 95% confidence interval (95% CI) obtained from the models were reported. *p* ≤ 0.05 is statistically significant, and IBM SPSS Statistics for Windows (version 23.0; IBM Corporation., New York, NY, USA) was used for all statistical analyses.

## 3. Results

12,641 adolescents (50.5%, 6388 boys; 49.5%, 6253 girls) were included in the final statistical analysis of this study. Descriptive statistical analysis results based on gender (Table 1) show that the average age of the adolescents in this study (40.2% urban, 59.8% rural; 67.2% low SES, 25.8% moderate SES, 7% high SES) was 15.04 years old, wherein 17.5% of the adolescents had tried cigarette smoking and 8.9% of the adolescents had smoked in the past 30 days. The results indicate that the proportion of age of first time cigarette smoking are: age 10 and under (8%), age 11–14 (5.4%), age 15 and above (4.1%). The percentages of smoking frequency are: occasional smokers (5.88%) and regular smokers (2.99%), and there is a significantly higher percentage of boys (occasional smokers: 7.06%; regular smokers: 3.93%) compared to girls (occasional smokers: 4.67%; regular smokers: 2.03%) (ꭓ^2^ = 75.17; *p* = 0.00; ES = 0.15). Moreover, PA score was also higher in boys at 2.34 compared to 2.18 in girls (*p* = 0.00; ES = 0.22). The percentages of screen time category are low ST (86.73%) and high ST (13.27%), and there was a significant difference between girls (low ST: 90.02%; high ST: 9.98%) and boys (low ST: 83.50%; high ST: 16.50%) (ꭓ^2^ = 116.70; *p* = 0.00; ES = 0.10). The average QOL of adolescents was 140.31 points, and there was no significant difference between males and females in the total QOL.

According to the classification of variables, it can be seen from the total score for QOL (Table 2) that urban (Boys M = 141.9; Girls M = 142.48) is significantly higher (Boys effect size (ES) = 0.138; Girls (ES) = 0.177) than rural (Boys 139.11; Girls 138.99); with a rise in SES, the total QOL also increased significantly (Boys (ES) = 0.008; Girls (ES) = 0.007); age of first cigarette smoking shows significant differences in the total score of QOL (Boys (ES) = 0.096; Girls (ES) = 0.073); smoking status in the past 30 days shows significant differences in the total score of QOL (Boys (ES) = 0.061; Girls (ES) = 0.048); high PA (Boys M = 142.71; Girls M = 143.41) is significantly higher (Boys (ES) = 0.340; Girls (ES) = 0.362) than low PA (Boys M = 135.93; Girls M = 136.37); high ST (Boys M = 142.16; Girls M = 141.60) is significantly higher (Boys (ES) = −0.581; Girls (ES) = −0.645) than low ST (Boys M = 130.67; Girls M = 129.10).

As shown in Table 3, the relationship between demographic factors, smoking and QOL scores was analyzed by a multiple linear regression model. The model shows that: adolescent rural (except for physical and mental health and social and psychological function of boys) and older boys and girls have a significant negative correlation with all QOL scores in the binary analysis. Compared with participants with high SES, there is a significantly lower score of QOL in participants with low or moderate SES (except for physical and mental health). First-time smokers aged 10 and under, 11 to 14 years, or 15 years and over (except for living environments) had significantly lower quality of life scores than those who had never smoked. Compared with participants with no smoking history in the past 30 days, there is a significantly lower score of QOL (except for physiological and mental health) in participants who are occasional smokers or regular smokers in the past 30 days (except for living environment of girls).

As shown in Table 4, through the multiple linear regression model, the analysis of categorical variables based on PA and ST shows that high PA is significantly positively correlated with all QOL scores, while high ST is significantly negatively correlated with all QOL scores. The relationship is significant in the continuous variables of PA level and ST, and more than 30 min of PA five or more days a week is most strongly associated with overall QOL, with a significantly positive correlation.

The relationship between ST-PA combination and QOL was analyzed by linear regression model. According to the model (Table 5), compared with the low-PA-low ST combination in different patterns of ST-PA combination (as a reference), the correlation between the QOL and the high PA-low ST combination is the strongest, with a significantly positive correlation (Boys β = 8.317; Girls β = 8.780). The low PA-high ST combination (Boys β = −8.029; Girls β = −9.769) has the lowest QOL score and a significant negative correlation. The high PA-high ST combination (Boys β = 1.637; Girls β = 2.043) has a higher QOL score. In addition, we compared different combinations of over 30 min of PA a week-high/low ST with 0 days a week for more than 30 min of PA-low ST combination (as reference) and the results showed the strongest association between QOL and more than 30 min of PA five days or more a week–low ST combination (Boys β = 13.144; Girls β = 13.130), with a significantly positive correlation. More than 30 min of PA on 0 days a week-high ST combination (Boys β = −6.773; Girls β = −6.801) had the lowest QOL scores, with a significantly negative correlation. In addition, boys who PA for more than 30 min five days a week with-high ST (β = 5.951) and girls who PA for more than 30 min three to four days a week-high ST (β = 3.699) scored higher on QOL.

## 4. Discussion

This study explores the relationship between QOL and smoking, PA, and ST among Chinese adolescents. The relationship between PA–ST combination and QOL and its relationship between quantity and effect were also explored, while accounting for demographic factors such as age, sex, place of residence and socioeconomic status. We found that adolescents with high levels of PA have a better QOL, and more than 30 min of PA 5 days and above per week are most strongly associated with a better QOL; older, rural, smoking and ST were associated with poorer QOL and those first trying cigarette smoking at the age of 10 and under had the worst QOL; in addition, more than 30 min of PA five or more days a week–low ST combination had the best QOL. Boys who did more than 30 min of PA five days a week and girls who did more than 30 min of PA three to four days a week had a better QOL than girls who did not do more than 30 min of PA on any day.

Chinese adolescents need a lot of time to study and prepare for exams, which causes some mental health problems [49,50], and high school adolescents have been under the most stress, which correlates with the result of this study showing a significant negative correlation between age and QOL. Previous studies have shown that environmental factors such as material deprivation and reduced social support are related to adolescent depression [37] and parents have a great influence on the entire family and children [51]. Limited family material resources and lower social status also hurt QOL [10], which is consistent with the results of this study. The quality of urban adolescents’ QOL is clearly better and adolescents with higher social status enjoy better QOL. In addition, girls are more vulnerable than boys [46], which may explain that we found the social and psychological function for urban girls is better than that for rural girls while there is no significant difference between boys. Therefore, when formulating strategies to improve the QOL of adolescents in the future, older, rural, and low SES of adolescents should be focused upon.

Our findings suggest that adolescents who smoke have a lower quality of life, compared to the behavioral health of (non-smoking) adolescents reported in previous studies, with general mental health and fewer depressive symptoms [52]. However, current studies have found no significant difference in quality of life between smokers and non-smokers [53]. We suspect that most of the samples in this study may be light smokers, or results may also be related to local perceptions of adolescent smoking. Nevertheless, smoking in China is a type of adult behavior [30], and teenagers smoking runs counter to Chinese collectivism and the cultural values of filial piety. Besides, smoking among young people is more likely to produce depression, suicidal thoughts, etc. [54], and smoking will harm health and increase the risk of chronic diseases in adulthood [55]. However, smoking behavior during adolescence is common [56]. Further, we found that Chinese adolescents aged 10 and under comprise the highest percentage of the population trying cigarettes, with the lowest quality of life. Therefore, when formulating strategies to improve the QOL of adolescents in the future, prohibiting smoking behavior may be the priority of the emphasis placed on adolescents who are aged 10 and under.

At present, there is a divergence in the relationship between PA and QOL of adolescents [27,34]. The results of this study support the positive correlation between PA and QOL of adolescents, which is shown by previous studies in which PA can significantly improve QOL [57,58], and related to well-being, life satisfaction and other aspects [59], which is applicable to the youth population [11]. This study also found that more than 30 min of PA 5 days a week and above is associated with better QOL among adolescents, which is consistent with the guidelines issued by the World Health Organization for young people to perform at least 60 min of moderate-intensity PA every day [60]. In fact, regular PA has additional benefits, such as improving cardiopulmonary function [61] and reducing the risk of various chronic diseases (respiratory problems, allergies, cancer, etc.) [62]. The current relationship between adolescents’ QOL and ST is controversial [12,35,36]. The results of this study support the apparent negative correlation between ST and QOL. This is consistent with previous research reports that adolescents’ ST is negatively correlated with life satisfaction [63], physical health condition [64], mental health condition [28] and other aspects of QOL. This may be because the time spent on watching TV has replaced the time spent on social activities and physical exercise [64]. In addition, TV content may also affect mental health. For example, watching violent programs may cause fear, anxiety or antisocial behavior [65]. Studies also show that the high ST of adolescents is associated with various diseases [25]. Therefore, the ST and PA of young people should be regulated as much as possible.

From the perspective of the relationship between PA–ST combination and QOL, previous studies have shown that increasing PA is more important for improving the health of adolescents than reducing ST [66], and high PA can reduce the adverse reactions of high ST [27]. This is consistent with the results of this study showing that high PA can improve the negative impact of high ST on QOL and improve QOL. Although there are studies that contradict this conclusion [67], this study’s ranking is based on the β coefficient of ST and PA which means that the forms and units of the variables are different and should not be compared with each other. Besides, this study initially explored the dose–response relationship and found that adolescents with more than 30 min of physical exercise 5 days or more per week and low ST have the best QOL. PA for boys more than 5 days a week and more than 30 min for girls 3–4 days a week can improve the negative impact of high ST on QOL and improve QOL. This difference between men and women may be related to differences in social and cultural roles, psychological attributes, and hormonal factors [68,69]. However, as far as we know, this study is the first to explore the dose–response relationship between PA–ST combination and QOL of adolescents. There is a lack of evidence to support this conclusion, and future high-quality research needs to be further verified.

Our research found that smoking behavior, PA level, ST duration, age, place of residence, and socioeconomic status of Chinese adolescents are all related to quality of life. In addition, our conclusion that high levels of PA could ameliorate the negative effects of high ST is consistent with previous studies [27]. Moreover, PA of over 30 min five days a week for boys and over 30 min three to four days a week for girls can effectively improve the negative effects of high ST. Therefore, school administrators should develop strategies based on gender differences to develop plans to promote PA levels, control adolescent smoking and high ST issues, and focus on older, rural and low-socioeconomic youth. This may effectively improve the quality of life of Chinese teenagers.

### Strengths and Limitations

This study has several strengths. First, the sample size is large and representative. Random large sampling allows us to better provide the required precision and avoid errors from testing a small number of possibly atypical samples. Second, measurement is by well-validated questionnaires, thus contributing to the reliability and validity of this approach. Third, the relationship between smoking, PA–ST combination and QOL in Chinese adolescents is discussed for the first time, and it is one of the only cross-sectional studies to date that has preliminarily explored the dose–response relationship of PA–ST combination. From this perspective, the study is significant because it provides a reference for formulating strategies to improve the QOL of adolescents in the future.

The limitations of the study include the following: First, this study is a cross-sectional study, which cannot accurately explain and analyze the cause and effect relationship. In the future, a longitudinal study is needed for further analysis. Second, this study has not explored other tobacco products such as cigarillo brands, hookah, and e-cigs, which are often more common than cigarettes. Third, the statistical analysis does not account for interactional effects. Fourth, the self-rated reports can bias results due to social desirability. Finally, this study did not distinguish the quality of screen time and the use of social media. We need to separate from the screen time that required by specific uses of social media in educational terms, such as professional study, health promotion, etc.

## 5. Conclusions

This study explored the relationship between smoking, PA, ST and QOL of Chinese adolescents, and explored the relationship between PA–ST combination and QOL, and the dose–response relationship, while accounting for demographic factors. We found that Chinese adolescents with high ST, smoking, older, rural, and lower SES had a poorer QOL, while those who first tried cigarette smoking at the age of 10 or under are most strongly associated with poorer QOL. A high level of PA is associated with a better QOL, and more than 30 min of PA five or more days a week was most strongly associated with better QOL. In addition, the group with high PA (more than 30 min of PA five or more days a week)–low ST had the best QOL. More than 30 min of PA five or more days a week for boys and three or four days a week for girls can improve the negative effects of high ST and improve QOL. Therefore, strategies for improving the QOL of Chinese adolescents should focus on smoking, PA, ST, the older, the rural and low SES adolescents and take into consideration the relationship between QOL and related PA–ST combinations.

## Figures and Tables

**Table 1 ijerph-17-08043-t001:** General characteristics of participants according to gender among adolescents.

Variables		Total (*n* = 12,641)	Boys (*n* = 6388)	Girls (*n* = 6253)	ES	*p*-Value
Age (years)		15.04 (1.70)	15.00 (1.69)	15.08 (1.71)	0.05	0.01
Place of residence, *n* (%)	Urban	5084 (40.22)	2636 (41.26)	2448 (39.15)		
	Rural	7557 (59.78)	3752 (58.74)	3805 (60.85)	0.02	0.02
Socioeconomic status, *n* (%)	Low	8489 (67.15)	4306 (67.41)	4183 (66.90)		
	Moderate	3265 (25.83)	1626 (25.45)	1639 (26.21)	0.01	0.58
	High	887 (7.02)	456 (7.14)	431 (6.89)		
Age when first tried cigarette smoking, *n* (%)	Never	10,415 (82.39)	4920 (77.02)	5495 (87.88)		
	Age 10 U	1017 (8.05)	638 (9.99)	379 (6.06)	0.15	
	Age 11–14	688 (5.44)	454 (7.11)	234 (3.74)		0.00
	Age 15 A	521 (4.12)	376 (5.89)	145 (2.32)		
Smoking status in the past 30 days, *n* (%)	NS	11,520 (91.13)	5686 (89.01)	5834 (93.30)		
	OS	743 (5.88)	451 (7.06)	292 (4.67)	0.08	0.00
	RS	378 (2.99)	251 (3.93)	127 (2.03)		
Physical activity score		2.26 (0.72)	2.34 (0.76)	2.18 (0.67)	0.22	0.00
Physical activity category, *n* (%)	Active	7632 (60.37)	4087 (63.98)	3545 (56.69)		
	Inactive	5009 (39.63)	2301 (36.02)	2708 (43.31)	0.07	0.00
Physical activity ≥ 30 min per week, *n* (%)	0 days	3584 (28.35)	1765 (27.63)	1819 (29.09)		
	1–2 days	2416 (19.11)	1228 (19.22)	1188 (19.00)		
	3–4 days	3238 (25.62)	1726 (27.02)	1512 (24.18)	0.04	0.00
	≥5 days	3403 (26.92)	1669 (26.13)	1734 (27.73)		
Screen time (hour)		0.66 (1.04)	0.78 (1.133)	0.53 (0.93)	0.24	0.00
Screen time category, *n* (%)	<2 h	10,963 (86.73)	5334 (83.50)	5629 (90.02)		
	≥2 h	1678 (13.27)	1054 (16.50)	624 (9.98)	0.10	0.00
	SPF QOL	61.16 (10.75)	60.77 (11.17)	61.55 (10.29)	0.07	0.00
	PMH QOL	34.93 (6.99)	35.01 (7.32)	34.85 (6.64)	0.02	0.20
QOL quality of life score	LE QOL	21.72 (4.68)	22.04 (4.82)	21.38 (4.51)	0.14	0.00
	Total QOL	140.31 (19.98)	140.26 (20.23)	140.36 (19.73)	0.02	0.78

Data were described as *n* (%) or mean ± SD; age 10 U, age 10 and under; age 15 A, age 15 and above; NS, never smoker, OS, occasional smoker; RS, regular smoker; SPF, social and psychological function; PMH, physiological and mental health; LE, living environment; ES, effect size.

**Table 2 ijerph-17-08043-t002:** Mean (SD) of total quality of life (QOL) component according to the level of demographic characteristics, screen time and physical activity among adolescents. Data are means (SD).

Variables		Boys (*n* = 6388)	Girls (*n* = 6253)
Place of residence	Rural	139.11 (19.56)	138.99 (18.90)
	Urban	141.90 (21.04)	142.48 (20.79)
	ES	0.14	0.18
	*p*-value	0.00	0.00
SES	Low	139.46 (19.35)	139.46 (19.14)
	Moderate	140.58 (21.72)	141.2 (20.72)
	High	146.7 (21.65)	145.81 (20.57)
	ES	0.03	0.02
	*p*-value	0.000	0.00
Age when first tried cigarette smoking	No	143.59 (19.88)	142.31 (19.36)
	Age 10 U	126 (16.22)	124.44 (15.89)
	Age 11–14	129.37 (17.37)	126.38 (15.67)
	Age 15 A	134.14 (17.38)	130.45 (17.89)
	ES	−0.10	−0.07
	*p*-value	0.000	0.000
Smoking status in the past 30 days	NS	142.02 (19.93)	141.52 (19.53)
	OS	125.81 (17.02)	124.13 (15.3)
	RS	126.41 (15.92)	124.32 (13.96)
	ES	−0.06	−0.05
	*p*-value	0.000	0.000
Physical activity	Inactive	135.93 (19.60)	136.37 (18.43)
	Active	142.71 (20.17)	143.41 (20.15)
	ES	0.34	0.36
	*p*-value	0.00	0.00
Physical activity ≥ 30 min per week	0 days	134.77 (20.35)	134.49 (18.93)
	1–2 days	137.67 (18.39)	138.51 (17.70)
	3–4 days	141.89 (19.22)	143.26 (19.41)
	≥5 days	147.24 (20.34)	147.25 (20.61)
	ES	0.06	0.06
	*p*-value	0.000	0.000
Screen time category	<2 h	142.16 (20.04)	141.60 (19.51)
	≥2 h	130.67 (18.35)	129.10 (18.15)
	ES	−0.58	−0.65
	*p*-value	0.00	0.00

SD, standard deviation; QOL, quality of life; SES, socioeconomic status; age 10 U, age 10 and under; age 15 A, age 15 and above; NS, never smoker, OS, occasional smoker; RS, regular smoker; ES, effect size.

**Table 3 ijerph-17-08043-t003:** Results from the multivariable general linear models evaluating the association of demographic factors and smoking with QOL in Boys and Girls.

Boys (*n* = 6388)				
Variables	SPF QOL	PMH QOL	LE QOL	Total QOL
	β (95% CI)	β (95% CI)	β (95% CI)	β (95% CI)
Age (year)	−1.155 (−1.307, −1.004) **	−0.751 (−0.856, −0.646) **	−0.166 (−0.232, −0.101) **	−1.379 (−1.645, −1.112) **
Place of residence (urban vs. rural)	0.027 (−0.507, 0.561)	0.237 (−0.133, 0.608)	0.865 (0.633, 1.097) **	1.166 (0.224, 2.109) *
SES (low vs. high)	−3.689 (−4.691, −2.687) **	−0.165 (−0.860, 0.531)	−2.106 (−2.540, −1.671) **	−6.824 (−8.592, −5.056) **
SES (moderate vs. high)	−3.282 (−4.337, −2.228) **	−0.418 (−1.150, 0.314)	−1.443 (−1.901, −0.986) **	−5.707 (−7.568, −3.846) **
Smoking				
AFTCM (year)	1.388 (1.052, 1.724) **	0.748 (0.516, 0.980) **	0.384 (0.238,0.529) **	3.091 (2.497, 3.685) **
AFTCM (age 10 U vs. no)	−5.567 (−6.500, −4.634) **	−2.164 (−2.812, −1.517) **	−1.814 (−2.219, −1.409) **	−11.926 (−13.572, −10.280) **
AFTCM (age 11–14 vs. no)	−3.919 (−4.990, −2.848) **	−2.304 (−3.047, −1.561) **	−1.147 (−1.611, −0.682) **	−8.866 (−10.755, −6.977) **
AFTCM (age 15 A vs. no)	−1.543 (−2.715, −0.371) *	−1.613 (−2.426, −0.800) **	−0.170 (−0.678,0.339)	−3.878 (−5.946, −1.811) **
SSP30D (number of cigarettes)	−0.058 (−0.090, −0.025) *	−0.007 (−0.029, 0.016)	−0.005 (−0.019,0.009)	−0.127 (−0.185, −0.070) **
SSP30D (occasional smoker vs. NS)	−4.170 (−5.329, −3.010) **	−0.283 (−1.088, 0.521)	−0.908 (−1.411,0.405) **	−7.865 (−9.911, −5.820) **
SSP30D (regular smoker vs. NS)	−5.007 (−6.412, −3.602) **	−0.158 (−1.133, 0.817)	−1.632 (−2.242, −1.022) **	−8.304 (−10.783, −5.825) **
Girls (*n* = 6253)				
Variables	SPF QOL	PMH QOL	LE QOL	Total QOL
	β (95%CI)	β (95% CI)	β (95% CI)	β (95% CI)
Age (year)	−1.064 (−1.201, −0.926) **	−0.895 (−0.987, −0.802) **	−0.194 (−0.255, −0.134) **	−1.360 (−1.618, −1.101) **
Place of residence (urban vs. rural)	1.052 (0.532, 1.573) **	0.193 (−0.157, 0.542)	1.377 (1.149, 1.605) **	2.943 (1.963, 3.922) **
SES (low vs. high)	−1.826 (−2.792, −0.860) **	−0.195 (−0.843, 0.453)	−1.539 (−1.961, −1.116) **	−4.242 (−6.059, −2.425) **
SES (moderate vs. high)	−1.642 (−2.642, −0.642) *	−0.053 (−0.724, 0.618)	−0.658 (−1.095, −0.221) *	−2.764 (−4.645, −0.883) *
Smoking				
AFTCM (year)	2.191 (1.757, 2.626) **	0.906 (0.614, 1.197) **	0.446 (0.257, 0.636) **	4.449 (3.631, 5.267) **
AFTCM (age 10 U vs. no)	−6.302 (−7.440, −5.164) **	−2.646 (−3.409, −1.882) **	−1.475 (−1.973, −0.978) **	−13.095 (−15.236, −10.954) **
AFTCM (age 11–14 vs. never)	−4.700 (−6.142, −3.259) **	−2.351 (−3.319, −1.384) **	−0.695 (−1.325, −0.065) *	−9.381 (−12.093, −6.670) **
AFTCM (age 15 O vs. no)	−4.048 (−5.695, −2.402) **	−1.636 (−2.741, −0.531) *	−0.548 (−1.268, 0.171)	−7.799 (−10.896, −4.702) **
SSP30D (number of cigarettes)	−0.082 (−0.122, −0.042) **	0.016 (−0.011, 0.043)	− 0.031 (−0.048, −0.013) **	−0.154 (−0.229, −0.079) **
SSP30D (occasional smoker vs. NS)	−4.847 (−6.262, −3.432) **	−1.003 (−1.952, 0.054)	−1.046 (−1.664, −0.427) *	−7.875 (−10.536, −5.214) **
SSP30D (regular smoker vs. NS)	−4.720 (−6.561, −2.880) **	−0.673 (−1.908, 0.562)	−0.640 (−1.445, 0.165)	−8.519 (−11.981, −5.057) **

β (95%CI), Regression coefficients and 95% confidence intervals; QOL, quality of life; SPF, social and psychological function; PMH, physiological and mental health; LE, living environment; SES, socioeconomic status; AFTCM, age when first tried cigarette smoking; age 10 U, age 10 and under; age 15 A, age 15 and above; SSP30D, smoking status in the past 30 days; NS, Never smoker; * 0.05, ** 0.01. The model was adjusted for physical activity and screen time. The variance inflation factors (VIF) values did not provide evidence of multicollinearity among the predictor variables (VIF < 10), therefore, the order of factors in the model is not explained.

**Table 4 ijerph-17-08043-t004:** Results from the multivariable general linear models evaluating the association of Physical activity and screen time with QOL in Boys and Girls.

Boys (*n* = 6388)				
Variables	SPF QOL	PMH QOL	LE QOL	Total QOL
	β (95% CI)	β (95% CI)	β (95% CI)	β (95% CI)
PA				
PA score	3.752 (3.417, 4.087) **	0.862 (0.623, 1.101) **	2.403 (2.261, 2.545) **	6.276 (5.681, 6.870) **
PA (active vs. inactive)	3.395 (2.858, 3.931) **	1.895 (1.522, 2.267) **	2.512 (2.279, 2.744) **	6.640 (5.693, 7.586) **
PA ≥ 30 min (1–2 days vs. 0 days) per week	1.659 (0.978, 2.340) **	1.363 (0.890, 1.836) **	−0.075 (−0.370, 0.221)	4.316 (3.115, 5.518) **
PA ≥ 30 min (3–4 days vs. 0 days) per week	2.747 (2.000, 3.495) **	1.718 (1.199, 2.237) **	0.553 (0.228, 0.877) *	7.198 (5.879, 8.517) **
PA ≥ 30 min (≥5 days vs. 0 days) per week	4.842 (4.155, 5.528) **	2.055 (1.578, 2.531) **	1.814 (1.516, 2.112) **	11.522 (10.310, 12.734) **
Screen time				
Screen time (hour)	−1.294 (−1.536, −1.053) **	−0.590 (−0.758, −0.422) **	−0.421 (−0.526, −0.317) **	−3.192 (−3.616, −2.767) **
Screen time category (high vs. low)	−3.003 (−3.723, −2.283) **	−1.364 (−1.864, −0.865) **	−0.939 (−1.251, −0.626) **	−7.430 (−8.700, −6.160) **
Girls (*n* = 6253)				
Variables	SPF QOL	PMH QOL	LE QOL	Total QOL
	β (95% CI)	β (95%CI)	β (95% CI)	β (95% CI)
PA				
PA score	3.099 (2.736, 3.462) **	−0.010 (−0.256, 0.236)	2.399 (2.244, 2.554) **	6.814 (6.132, 7.496) **
PA (active vs. inactive)	3.084 (2.594, 3.573) **	0.378 (0.049, 0.706) *	2.342 (2.128, 2.556) **	7.124 (6.203, 8.045) **
PA ≥ 30 min (1–2 days vs. 0 days) per week	1.615 (0.994, 2.235) **	1.605 (1.188, 2.021) **	0.078 (−0.193, 0.349)	3.949 (2.781, 5.117) **
PA ≥ 30 min (3–4 days vs. 0 days) per week	3.061 (2.362, 3.761) **	2.085 (1.616, 2.555) **	0.848 (0.542, 1.154) **	7.512 (6.196, 8.827) **
PA ≥ 30 min (≥5 days vs. 0 days) per week	4.416 (3.761, 5.070) **	2.582 (2.143, 3.021) **	1.737 (1.451, 2.024) **	10.989 (9.758, 12.221) **
Screen time				
Screen time (hour)	−1.411 (−1.693, −1.130) **	−1.015 (−1.204, −0.825) **	−0.530 (−0.653, −0.407) **	−3.720 (−4.248, −3.192) **
Screen time category (high vs. low)	−2.845 (−3.693, −1.996) **	−2.211 (−2.780, −1.642) **	−1.174 (−1.545, −0.803) **	−7.787 (−9383, −6.191) **

β (95% CI), Regression coefficients and 95% confidence intervals; PA, physical activity; QOL, quality of life; SPF, social and psychological function; PMH, physiological and mental health; LE, living environment; * 0.05, ** 0.01. The model was adjusted for age, place of residence, socioeconomic status and smoking, physical activity (regarding screen time) and screen time (regarding physical activity). The VIF values did not provide evidence of multicollinearity among the predictor variables (VIF < 10), therefore, the order of factors in the model is not explained.

**Table 5 ijerph-17-08043-t005:** Results from the multivariable general linear models evaluating the joint association of physical activity and screen time with total quality of life in adolescents in Boys and Girls.

Variables	Boys (*n* = 6388)	Girls (*n* = 6253)
	β (95% CI)	β (95% CI)
Low PA–Low ST	Reference	Reference
Low PA–High ST	−8.029 (−10.355, −5.704) **	−9.769 (−12.687, −6.850) **
High PA–Low ST	8.317 (7.299, 9.336) **	8.780 (7.824, 9.735) **
High PA–High ST	1.637 (0.001, 3.273) *	2.043 (0.082,4.004) *
PA ≥ 30 min 0 days per week–Low ST	Reference	Reference
PA ≥ 30 min 1–2 days per week–Low ST	4.258 (2.933, 5.584) **	4.536 (3.291, 5.780) **
PA ≥ 30 min 3–4 days per week–Low ST	8.690 (7.246, 10.134) **	9.144 (7.747, 10.541) **
PA ≥ 30 min ≥ 5 days per week–Low ST	13.144 (11.834, 14.453) **	13.130 (11.844, 14.416) **
PA ≥ 30 min 0 days per week-High ST	−6.773 (−9.320, −4.226) **	−6.801 (−9.831, −3.771) **
PA ≥ 30 min 1–2 days per week-High ST	−1.843 (−4.132, 0.447)	−2.114 (−4.973, 0.746)
PA ≥ 30 min 3–4 days per week-High ST	0.377 (−2.232, 2.985)	3.699 (0.685, 6.713) *
PA ≥ 30 min ≥ 5 days per week-High ST	5.951 (3.642, 8.261) **	3.184 (−0.073, 6.441)

β (95% CI), Regression coefficients and 95% confidence intervals; PA, physical activity; ST, screen time, * *p* < 0.05, ** *p* < 0.01. The model was adjusted for age, place of residence, socioeconomic status (SES) and smoking.

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
