# Peer review of "Relationship between Smoking, Physical Activity, Screen Time, and Quality of Life among Adolescents"

_ijerph, 2020, doi:10.3390/ijerph17218043_

Round 1

Reviewer 1 Report

The authors addressed most of my concerns. Good work

There is still a careful need to proofread this work because I could still find many styling problems.

Good luck.

Reviewer 2 Report

I have read through the revised manusript and also checked it against the suggested corrections/revisions. I am satisfied that the manusript has been revised adequately and it has improved greatly. I recommend for publication with no reservations.

Reviewer 3 Report

Many, more than enough, improvements have been incorporated.

This manuscript is a resubmission of an earlier submission. The following is a list of the peer review reports and author responses from that submission.

Round 1

Reviewer 1 Report

Dear Authors,

Thanks for giving me the chance to reread this revised manuscript, “Relationship between Smoking, Physical Activity, Screen Time, and Quality of Life among Adolescents”. Smoking indeed is a universal health problem, especially for teenagers all around the world and its effect on physical behavior is increasingly attracting academic attention. The revised version seems significantly improved. However, there are still several points worth noting which must be appropriately addressed.

  1. Insufficient literature background

The introduction part lacks the state of art smoking studies, which would definitely help your work to reach a broader audience. In line 45, you could add a sentence, “As smoking issues are attracting academic attention (Gao et al., 2020; Qin et al., 2019; Liu et al., 2020), it is the most urgent that the prevalence of smoking among people aged 15-24 years has gone up from 8.3% in 2003 (8.0-8.6 years) to 12.5% in 2013 (12.2-12.9 years)” to have a more grounded theory.

Reference:

Gao, L.; Gan, Y.; Whittal, A.; Lippke, S. Problematic Internet Use and Perceived Quality of Life: Findings from a Cross-Sectional Study Investigating Work-Time and Leisure-Time Internet Use. Int. J. Environ. Res. Public Health 2020, 17, 4056.

Qin, Z.; Song, Y.; Jin, Y. Green Worship: The Effects of Devotional and Behavioral Factors on Adopting Electronic Incense Products in Religious Practices. Int. J. Environ. Res. Public Health 2019, 16, 3618.

Liu, B.; Sze, J.; Li, L.; Ornstein, K.A.; Taioli, E. Bivariate Spatial Pattern between Smoking Prevalence and Lung Cancer Screening in US Counties. Int. J. Environ. Res. Public Health 2020, 17, 3383.

  1. Problems in presentation, methods, and analysis

The authors are advised to provide a better illustration of the table. In Table 5, it seems rare to use confidence interval, rather than the standard deviation, to illustrate the distribution of variables. Still, authors are also welcomed to share their data for further statistical validation.

Although I personally like this paper and its contributions, still problems must be addressed in order to further proceed. Hope these suggestions help.

Reviewer 2 Report

Overall this is a good paper and the authors have good justification for the study. The following suggestions will contribute to improving your paper. 

1). In the abstract, line 12 the word 'examine' missing some letters

2). Line 41, instead of 'etc' it is better to use 'and more' or 'among others' which are more formal in writing.

3). (i) It is confusing and erronous where the authors have written on Lines 91 - 92 12,900 young people (aged 11- 18 years) participated in the study. Then immediately after this, the next sentence says 259 adolecents were excluded from the study. How could the 259 have participated when they were excluded? This sentence should state 12,641 young people (and not 12,900) participated in the study.

(ii) May be 12,900 were sampled but 12,641 participated as seen in the results.

4). From lines 158 - 163 this is a very long and loaded sentence. It is hard for the reader to keep up with the flow of the information. Break up into two sentences.

5). Line 244 - The grammar is not right where you state "did more 30 minutes of PA on zero days". Could be reframed as "did not do more than30 minutes of PA on any day".

6). Line 269 - 271 Why is it only the priority for school management to prohibit smoking, why not include parents (family and friends) and also peer education? Does this imply more of the smoking is done in school? Need to justify why you make this assertion and not others.

7). Line 315 - How does the random large sampling help avoid errors and which particular errors? The reader needs to know what you are referring to.

8). Line 322 - Grammar is not right. May say 'limitations of the study include the following:  " instead of saying 'concluded in the followings'.

Finally, I think it is important to note that self-rated reports also bias results due to social desirability. This may be included as a limitation to the study.

Reviewer 3 Report

The study is meritorious. I apologize because I will now focus on the aspects to be improved, rather than on the more positive aspects of the report.

“investigate the dose-response 14 relationship between QOL and PA-ST combination” could be better “investigate the dose-response 14 relationship between PA-ST and QOL combination”, as PA-ST corresponds to the dose (cause) and QOL to response (consequence).

It is convenient to mention the data gathering tools in the abstract.

There are problems of gender matching, of punctuation marks, capital letters in inappropriate places, problems of concordance between plurals and singulars... Language must be improved.

31 - Clarify the relation between health and QOL concepts, specially if one of them includes the other.

63 – “the relationship between PA, ST and health condition is still unclear 63 [12]. Therefore, we need to formulate corresponding prevention and improvement strategies, and 64 targeted the differences on PA, ST and QOL between male and female teenagers.” Relation knowledge is needed for formulating strategies, not unclear knowledge.

68 – “Results are mixed since students achieve recommended physical activity 68 indicators and relationships with quality of life”. The sentence is unclear.

87 – It is convenient to describe the population size (schools and students), and to estimate the level of representativity of the sample. I miss some reference to stratification in population and sample.

133 – Reliability of subscale scores should be reported.

141 – There is just a very short reference to the validity of the QOL scale. It is convenient to justify or explain more the validity of the scale.

145 – t-test is used for comparing two group means (not ≤ 2).

163 – “There was also a significant difference in the proportion of girls and boys (boy>girl)”. Language needs to be improved. Include reference to chi squared, p-value and some effect size indication. The same in the next sentence.

In Table 1  “ES” I don’t understand the meaning of one single calculation for 3 categories (e.g. 0,009 for 3 socioeconomic status). There are 1 decimal, 2 decimal and 3 decimal numbers. Unify.

171 – it is convenient to comment effect sizes.

Table 2 is confusing to read. Too much information. It is convenient to prioritize the most important. No references to any statistic and/or effect size.

195 – It is convenient to explain more the reasons for separating regressions in boys and girls, instead of introducing gender as one more independent variable; and for splitting the analysis of demographic and physical activity factor effects.

It is convenient to clarify if the reported betas are standardized or not. In this case, having diverse and non-familiar metrics, it is more convenient to report standardized coefficients.

It is convenient to report the potential presence of collinearity effects. If they are significative, then the order of factors in the model should be explained or justified.

199 – “The model was adjusted for age (regarding place of residence, socioeconomic status and smoking), place of residence (regarding age, socioeconomic status and smoking), socioeconomic status (regarding age, regarding place of residence and smoking), smoking (regarding age, place of residence and socioeconomic status), physical activity and screen time.” is a confusing sentence.

307 – Causality links are not demonstrated in this kind of study designs, therefore it is convenient to moderate the statements. Like “our conclusion that high levels of PA [COULD] ameliorate the negative effects of high ST is consistent with previous studies.”.